# Characteristics, Comparative Analysis, and Phylogenetic Relationships of Chloroplast Genomes of Cultivars and Wild Relatives of Eggplant (*Solanum melongena*)

**Qihong Yang** [1,†]**, Ye Li** [2,†]**, Liangyu Cai** [1]**, Guiyun Gan** [1]**, Peng Wang** [1]**, Weiliu Li** [1]**, Wenjia Li** [1]**, Yaqin Jiang** [1]**, Dandan Li** [1]**, Mila Wang** [1,3]**, Cheng Xiong** [4]**, Riyuan Chen** [1,3] **and Yikui Wang** [1,*]

1 Institute of Vegetable Research, Guangxi Academy of Agricultural Sciences, Nanning 530003, China
2 Habin Academy of Agricultural Sciences, Harbin 150008, China
3 College of Horticulture, South China Agricultural University, Guangzhou 510642, China
4 Engineering Research Center for Horticultural Crop Germplasm Creation and New Variety Breeding, Ministry of Education, Changsha 410128, China
* Correspondence: ykwang@gxaas.net; Tel.: +86-771-3186-372
† These authors contributed equally to this work.

**Abstract:** The eggplant (*Solanum melongena*) is a popular vegetable around the world. However, the origin and evolution of eggplant has long been considered complex and unclear, which has become the barrier to improvements in eggplant breeding. Sequencing and comparative analyses of 13 complete chloroplast (cp) genomes of seven *Solanum* species were performed. Genome sizes were between 154,942 and 156,004 bp, the smallest genome was from *S. torvum* and the largest from *S. macrocapon*. Thirteen cp genomes showed highly conserved sequences and GC contents, particularly at the subgenus level. All genes in the 13 genomes were annotated. The cp genomes in this study comprised 130 genes (i.e., 80 protein-coding genes, 8 rRNA genes, and 42 tRNA genes), apart from *S. sisymbriifolium*, which had 129 (79 protein-coding genes, 8 rRNA genes, and 42 tRNA genes.). The *rps16* was absent from the cp genome of *S. sisymbriifolium*, resulting in a nonsense mutation. Twelve hotspot regions of the cp genome were identified, which showed a series of sequence variations and differed significantly in the inverted repeat/single-copy boundary regions. Furthermore, phylogenetic analysis was conducted using 46 cp genomic sequences to determine interspecific genetic and phylogenetic relationships in *Solanum* species. All species formed two branches, one of which contained all cultivars of the subgenus *Leptostemonum*. The cp genome data and phylogenetic analysis provides molecular evidence revealing the origin and evolutionary relationships of *S. melongena* and its wild relatives. Our findings suggest precise intra- and interspecies relatedness within the subgenus *Leptostemonum*, which has positive implications for work on improvements in eggplant breeding, particularly in producing heterosis, expanding the source of species variation, and breeding new varieties.

**Keywords:** *Solanum melongena*; chloroplast genome; DNA barcode; comparative analysis; phylogenetic; interspecific hybridization

## 1. Introduction

Solanaceae plants are medium-sized angiosperms, the largest group of vegetable crops, and the third-largest group of economic plants [1]. The taxa of Solanaceae plants are abundant and diverse, containing 90 genera and including 3000–4000 species. This genus includes many important crop species, raw industrial materials, and certain plant models used in research. Therefore, *Solanaceae* plants are often regarded as important research materials [2]. Due to the considerable economic and research significance of the *Solanaceae* family, taxonomic and phylogenetic studies on this family have received much attention [3–5]. The genus *Solanum* accounts for half of the Solanaceae family,

with approximately 2000 species that are mostly distributed in tropical and subtropical regions [1,6]. Wild species of *Solanum* are found in Southeast Asia, Africa, South America, and southern China. *Solanum* is large and strongly monophyletic, which makes taxonomic analysis of this group difficult [1,6,7]. Isolation and differentiation of species are the basic processes involved in new species formation. However, there is no strict reproductive isolation in *Solanum* plants. Their pollen exhibits a certain affinity, and many species have the potential to cross-hybridize [8,9]. Further, during the process of artificial hybridization and variety selection, many intermediate hybrids are also produced. Therefore, there are many controversies regarding the identities of local breeds, intermediate hybrids, and related wild species, as well as their evolutionary relationships and classifications [10,11].

The chloroplast (cp) genome is the most prominent marker of green cells in leaves and occupies the second-largest area in mesophyll cells, besides the vacuole [12]. Plastids have diverse functions, including starch biosynthesis, nitrogen metabolism, sulfate reduction, fatty acid synthesis, and DNA and RNA synthesis [13]. The cp genomes of most terrestrial plants are circular chromosomes consisting of four parts: two inverted repeats (IR), a large single copy (LSC) region, and a small single copy (SSC) region [14]. In photosynthetic organisms, the genome length is 115–165 kb [14,15]. Most cp DNAs shows monoparental inheritance, with relatively few recombination and intraspecific mutation events. The phylogenetic tree can be constructed without relying on any other data, similar to investigations of the evolutionary history of plants [3,16]. The taxonomic and phylogenetic studies of the subgenus *Leptostemonum* have always been an area of great interest, and the species-level taxonomy of three cultivated aubergine species, the brinjal eggplant (*Solanum melongena*), the scarlet eggplant (*S. aethiopicum*) and the gboma eggplant (*S. macrocarpon*), which has long been considered complex [1,3,17–19].

With the rapid development of sequencing technology, bringing genomics into the large-scale, low-cost, high-throughput sequencing era has greatly promoted the research progress of the chloroplast genome in plants. The cp genomes of more than 1800 plants and 94 *Solanum* plants are available from the National Centre for Biotechnology Information (NCBI) Organelle Genome Resources. These cp genomes are widely used in comparative genomics and research on plant phylogeny and classification. The subgenus *Leptostemonum* contains a wide range of plants; however, its classification remains unclear.

In this study, we utilized Illumina HiSeq and analyzed 13 cp genomes of 7 *Leptostemonum* species. A phylogenetic tree was constructed to assess the interspecific differences and structural patterns of the cp genomes within the genus. Based on these results, the origin and evolutionary relationships of *Leptostemonum* species were analyzed to reveal the genetic resources available for use in eggplant breeding.

## 2. Materials and Methods

### 2.1. Materials

Seven *Leptostemonum* species were collected from Africa, South America, Southeast Asia, and south China and transplanted in the Resource Garden of Vegetable Research Institute, Guangxi Academy of Agricultural Science, Nanning, China. Thirteen samples of the seven *Leptostemonum* species were obtained for cp genome sequencing.

### 2.2. DNA Extraction, Genome Sequencing, and Assembly

Approximately 5 g of fresh leaves was harvested for cp DNA isolation using an improved extraction method [20]. After DNA isolation, 1 μg of purified DNA was fragmented to construct short-insert libraries (insert size 430 bp) according to the manufacturer's instructions (Illumina), followed by sequencing on the Illumina HiSeq 4000 [21]. Prior to assembly, low-quality raw reads with adaptors showing a quality score below 20 or containing N were filtered. The cp genome was reconstructed using a combination of de novo and reference-guided assemblies, and the following three steps were used to assemble the cp genomes [22]. First, filtered reads were assembled into contigs using SOAPdenovo2.04 [23]. Second, contigs were aligned to the reference genome of eggplant using

BLAST and aligned contigs (≥80% similarity and query coverage) were ordered according to the reference genome. Third, clean reads were mapped to the assembled draft cp genome to correct wrong bases, and the majority of gaps were filled through local assembly by GapCloser. Insertions/deletions (InDels) of 13 cp genomes were assessed. The physical map of the circular cp genome of the 13 samples were drawn by Organellar GenomeDRAW (http://ogdraw.mpimp-golm.mpg.de/cgi-bin/ogdraw.pl (accessed on 26 March 2019)).

### 2.3. Genome Annotation

The cp protein-coding, transfer RNA (tRNA), and ribosomal RNA (rRNA) genes were annotated using the online DOGMA tool using default parameters [24]. A whole cp genome BLAST search (E-value $\leq 10^{-5}$, minimal alignment length percentage $\geq 40\%$) was performed against five databases: the Kyoto Encyclopedia of Genes and Genomes, Clusters of Orthologous Groups, Non-Redundant Protein Database databases, Swiss-Pro, and ChloroplastDB [25–28]. The circular chloroplast genome map of *Solanum* was drawn using Organellar Genome DRAW v1.2. The cp genomes of the 13 samples were compared using VISTA [29]. Genome, protein-coding gene, intron, and spacer sequence divergences were evaluated using DnaSP 5.10 after alignment [30]. The genomic sequences were aligned using MAFFT v5 and adjusted manually where necessary [31]. For protein-coding gene sequences, introns, and spacers, every gene or fragment was edited using the ClustalW multiple alignment option within BioEdit v7.0.9.0 [32].

### 2.4. IR Expansion and Contraction

IR expansion and contraction was determined using the site https://irscope.shinyapps.io/irapp/analysis (accessed on 26 March 2019).

### 2.5. Codon Usage

The EMBOSS-6.6.0 cusp module was used to analyze the coding sequence area [33].

### 2.6. Comparative Genome Analysis

MUMmer (3.23) and BLAST were used to conduct global and local alignments between the sample and reference genomes, determining potential single-nucleotide polymorphisms (SNPs). Subsequently, SNPs were filtered out in the repeat regions as detected using the software BLAST, Repeat Masker, and TRF. Finally, SNPs were annotated based on the position and interaction between genes. LASTZ software (Release 1.04.15) was used for global alignment between each sample sequence and the reference genome. The alignment result was corrected using axt_correction, axtSort, and axtBest to determine potential InDels with lengths less than 50 bp. Finally, BWA and SAM tools were used to map the reads to InDel sequences and filter out unreliable InDels [34,35].

### 2.7. Repeat and SSR Analysis

Repeats and simple sequence repeats (SSRs) were determined using the website https://bibiserv.cebitec.uni-bielefeld.de/reputer (accessed on 26 March 2019). Analysis, parameters of minimum repeat length 30 bp and hamming distance 3. SSRs are found widely in the genome. Generally, they are composed of 1–6 bp repeat sequences with a low degree, with 2–3 nucleotides as repeat units such as (GA)n, (AC)n, and (GAA)n. The MIcroSAtellite identification tool (MISA) was used to detect microsatellite loci. Definition (unit_size, min_repeats): 1–10, 2–6, 3–5, 4–5, 5–5, 6–5. The minimum distance between the two SSRs was set to 100 bp. Parameter meaning: 1 base repeat 10 time or more; 2 bases repeat 6 times or more; 3 base repeats 5 times or more; repeat 4 bases for 5 times or more; 5 base repeats 5 times or more; a sequence with 6 base repeats of 5 or more times is considered a microsatellite sequence. Concurrently, when the distance between the two microsatellites was less than 100 bp, they formed a composite microsatellite.

*2.8. Identification of the Most Variable Regions*

After aligning the sequence, the Pi value was calculated using a sliding window (window 300 bp, step size 200 bp). A window with a *Pi* value greater than or equal to 0.01 was selected and primers were designed at both ends of the window sequence.

*2.9. Phylogenetic Analysis*

ClustalW was used to align the cp DNA sequences with default parameters, and the alignment was checked manually [32]. Maximum-likelihood (ML) methods were used for gnome-wide phylogenetic analyses using PhyML 3.0 [36]. The nucleotide substitution model was selected using jModelTest 2.1.10 and Smart Model Selection in PhyML 3.0 [37]. TheGeneral Time Reversible model (GTR model) with default parameter was selected for ML analyses with 1000 bootstrap replicates to calculate the bootstrap values of the topology. The results were treated with iTOL 3.4.3 [38].

**3. Results**

*3.1. Characteristics of Cp Genomes of Leptostemonum Species*

The 13 cp DNA samples of seven *Leptostemonum* species were sequenced using the Illumina HiSeq 4000 (San Diego, CA, USA): 5120–10,165 Mb raw paired-end reads were generated, with an average organelle depth ranging from 636 to 3639 and a Q30 of 92.27–98.72% (Table 1). Data from the 13 complete cp genomes were submitted to NCBI (Table 2). All 13 complete cp genomes consisted of a single circular double-stranded DNA molecule with a classic four-part structure, as observed in most angiosperms, including the LSC, SSC, IRA, and IRB regions (Figure 1).

No prominent sequence inversions of genomic rearrangements were observed. Among the 13 cp genomes, the assembled length was 154,942–156,004 bp, LSC length was 85,646–86,542 bp, SSC length was 18,435–18,602 bp, IRA and IRB lengths were consistently 25,420–25,461 bp, and total GC content was 37.76–37.81% (Table 3). These results indicated that there were no significant differences in the genome size, GC content, LSC length, SSC length, IRA length, and IRB length of the seven *Solanum* species. These results also demonstrated the characteristics of *Solanum* cp genomes, which included short-length and conserved genes and genomic structures. Most *Solanum* species are shown together in a pie chart, as their numbers, orders, and gene names were the same (130 genes, i.e., 80 protein-coding genes, 8 rRNA genes, and 42 tRNA genes) (Figure 1A, Tables 3 and 4). NN15 (*Solanum sisymbriifolium*) was annotated in a separate pie chart as it only contained 129 genes (79 protein-coding genes, 8 rRNA genes, and 42 tRNA genes), and the *rps16* gene was divided into two parts in NN15 that resulted in a nonsense mutation (Figure 1B, Tables 3 and 4).

**Table 1.** List of test materials.

| Sample Number | Code | Species | Gene Bank Accession Number | Origin |
|---|---|---|---|---|
| NN1 | 166 | *Solanum aethiopicum* L. | MN218076 | South Africa |
| NN2 | 53 | *Solanum aethiopicum* L. | MN218077 | Brazil |
| NN3 | 137 | *Solanum aethiopicum* L. | *MN218078* | Ethiopia |
| NN4 | Y11 | *Solanum aethiopicum* L. | *MN218079* | Ethiopia |
| NN5 | 177 | *Solanum melongena* L. | MN218080 | South China |
| NN6 | Lwpq | *Solanum macrocapon* L. | MN218081 | Laos |
| NN7 | Shf | *Solanum melongena* L. | *MN218082* | Viet Nam |
| NN8 | 131 | *Solanum melongena* L. | *MN218083* | South China |
| NN9 | Dhq | *Solanum wrightii* L. | MN218084 | South China |
| NN12 | Sq | *Solanum torvum* | MN218087 | South China |
| NN13 | Ctq | *Solanum anguivi* L. | *MN218088* | Thailand |
| NN14 | ctq-B | *Solanum anguivi* L. | *MN218089* | South China |
| NN15 | Sjq | *Solanum sisymbriifolium* | MN218090 | South China |

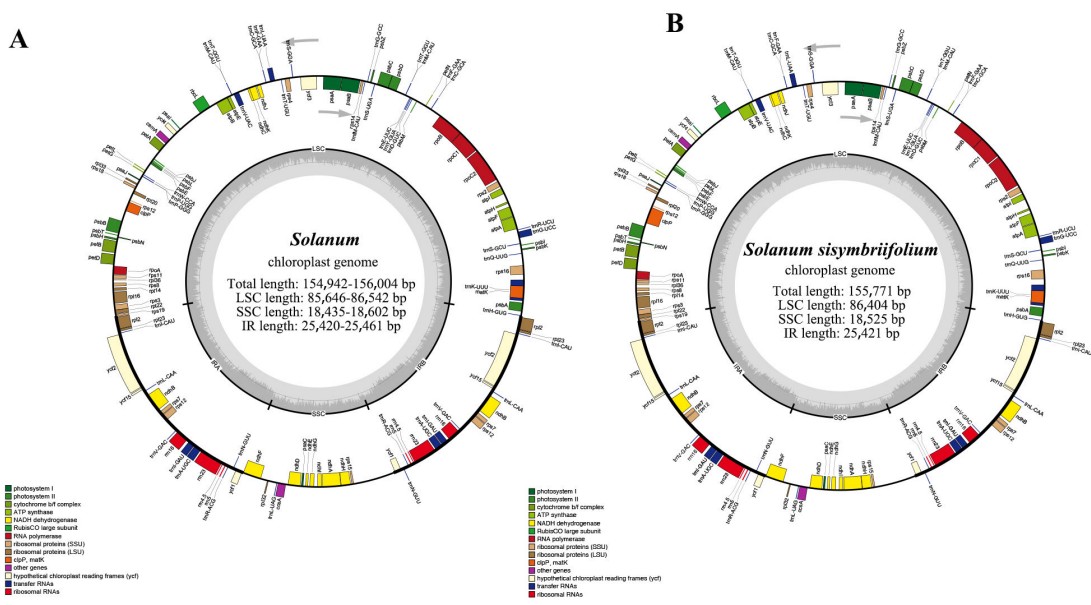

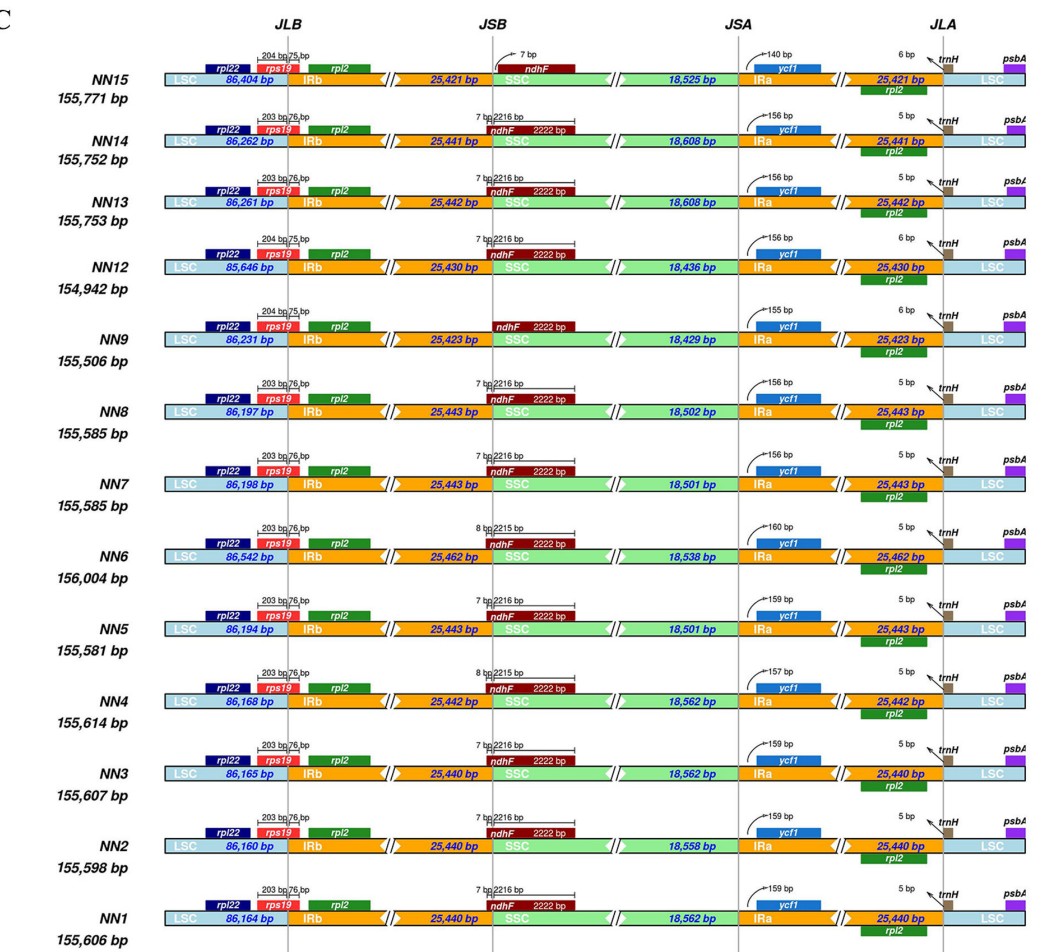

**Figure 1.** Structure of 13 cp genomes. (**A**) The circle characteristic of the reference genome cp genomes (*S. melongena*). (**B**) The physical map of the circular cp genome of *S. sisymbriifolium*. Genes shown inside the outer circle are transcribed counterclockwise, and those outside are transcribed clockwise. Genes belonging to different functional groups are color-coded. A gray area in the inner circle indicates the GC content. (**C**) Comparison of the borders of large single-copy (LSC), small single-copy (SSC), and inverted repeat (IR) regions among the cp genomes of 13 cp genomes.

**Table 2.** Summary of sequencing data for 13 cp genomes.

| Sample ID | Insert Size (bp) | Raw Data (Mb) | Clean Data (Mb) | Reads Length (bp) | Clean Data GC(%) | Clean Data Q30(%) | Average Organelle Depth |
|---|---|---|---|---|---|---|---|
| NN1 | 430 | 5981 | 5210 | (150:150) | 36.65 | 93.81 | 1452 |
| NN2 | 430 | 9111 | 8248 | (150:150) | 36.36 | 94.4 | 1521 |
| NN3 | 430 | 6754 | 6034 | (150:150) | 36.47 | 94.21 | 1336 |
| NN4 | 430 | 6374 | 5638 | (150:150) | 36.91 | 93.95 | 1824 |
| NN5 | 430 | 6515 | 5792 | (150:150) | 36.48 | 94.13 | 1599 |
| NN6 | 430 | 8670 | 8302 | (150:150) | 36.92 | 95.39 | 1477 |
| NN7 | 430 | 8584 | 8307 | (150:150) | 37.64 | 95.74 | 1581 |
| NN8 | 430 | 9893 | 9557 | (150:150) | 37.13 | 95.64 | 1798 |
| NN9 | 430 | 8504 | 8193 | (150:150) | 37.12 | 95.86 | 700 |
| NN12 | 430 | 10,165 | 9777 | (150:150) | 38.1 | 95.54 | 3639 |
| NN13 | 430 | 8408 | 8139 | (150:150) | 36.42 | 95.78 | 2337 |
| NN14 | 430 | 5781 | 5193 | (150:150) | 36.24 | 95.31 | 769 |
| NN15 | 430 | 5281 | 4503 | (150:150) | 39.27 | 93.5 | 636 |

**Table 3.** Information on 13 cp gene annotation.

| Sample ID | Total Length (bp) | LSC Length (bp) | SSC Length (bp) | IRA/IRB Length (bp) | Total GC Content (%) | Protein-Coding Gene Number | tRNA Number | rRNA Number | Total Gene Number |
|---|---|---|---|---|---|---|---|---|---|
| NN1 | 155,606 | 86,164 | 18,562 | 25,440 | 37.7 | 80 | 42 | 8 | 130 |
| NN2 | 155,598 | 86,160 | 18,558 | 25,440 | 37.7 | 80 | 42 | 8 | 130 |
| NN3 | 155,607 | 86,165 | 18,562 | 25,440 | 37.7 | 80 | 42 | 8 | 130 |
| NN4 | 155,614 | 86,168 | 18,564 | 25,441 | 37.7 | 80 | 42 | 8 | 130 |
| NN5 | 155,581 | 86,194 | 18,501 | 25,443 | 37.71 | 80 | 42 | 8 | 130 |
| NN6 | 156,004 | 86,542 | 18,538 | 25,462 | 37.61 | 80 | 42 | 8 | 130 |
| NN7 | 155,585 | 86,197 | 18,502 | 25,443 | 37.71 | 80 | 42 | 8 | 130 |
| NN8 | 155,585 | 86,197 | 18,502 | 25,443 | 37.71 | 80 | 42 | 8 | 130 |
| NN9 | 155,506 | 86,231 | 18,429 | 25,423 | 37.68 | 80 | 42 | 8 | 130 |
| NN12 | 154,942 | 85,646 | 18,436 | 25,430 | 37.81 | 80 | 42 | 8 | 130 |
| NN13 | 155,753 | 86,261 | 18,608 | 25,442 | 37.66 | 80 | 42 | 8 | 130 |
| NN14 | 155,752 | 86,262 | 18,608 | 25,441 | 37.67 | 80 | 42 | 8 | 130 |
| NN15 | 155,771 | 86,404 | 18,525 | 25,421 | 37.76 | 79 | 42 | 8 | 129 |

**Table 4.** List of genes encoded by 13 cp genomes.

| Function | Gene Group | Gene Name |
|---|---|---|
| Photosynthesis pathways | Photosystem I | *psa (A, B, C \*, I, J)* |
| | Photosystem I assembly | *ycf (3, 4)* |
| | Photosystem II | *psb (A-F, H-L, N, T, Z)* |
| | F-type ATP synthase | *atp (A, B, E, F, H, I)* |
| | NDH complex | *ndh (A \*, B #, C, D \*, E \*, F \*, G \*, H \*, I \*, J, K)* |
| | Component of cytochrome b6/f complex | *pet (A, B, D, L)* |
| | Inner envelope membrane | *cemA* |
| | Cytochrome biogenesis protein | *ccsA \** |
| | Large subunit of Rubisco | *rbcL* |
| Structural RNAs | Transfer RNAs | *trnH-GUG; trnK-UUU; trnQ-UUG; trnS-GCU; trnG-UCC; trnR-UCU; trnF-GAA; trnD-GUC; trnY-GUA; trnE-UUC; trnT-GGU; trnM-CAU; trnS-UGA; trnC-GCA; trnG-GCC; trnfM-CAU; trnS-GGA; trnT-UGU; trnL-UAA; trnI-CAU #; trnV-UAC; trnW-CCA; trnP-UGG; trnP-GGG; trnI-GAU #; trnA-UGC #; trnN-GUU#; trnL-UAG; trnR-ACG #; trnV-GAC #; trnL-CAA #* |
| | Ribosomal RNAs | *rrn (4.5 #, 5 #, 16 #, 23 #)* |

**Table 4.** *Cont.*

| Function | Gene Group | Gene Name |
|---|---|---|
| Genetic apparatus | Large subunit of ribosomal protein <br> Small subunit of ribosomal protein <br> Subunits of RNA polymerase | *rpl (2 [#], 14, 16, 20, 22, 23 [#], 32 \*, 33, 36)* <br> *rps (2, 3, 4, 7 [#], 8, 11, 12 [#&], 14, 15 \*, 16 [0], 18, 19)* <br> *rpo (A, B, C1, C2)* |
| Post-transcriptional modification | Maturase | *MatK* |
| Protein-modifying | ATP-dependent Clp protease proteolytic subunit | *clpP* |
| Unknown Proteins | | *ycf (1 [#], 2 [#], 15)* |

[#], Gene repeated in IR region; \* gene in SSC region; [&], gene has 2 separate transcription units; [0] gene is not in material NN15.

The expansion and contraction of IR regions can cause differences in the genome size. Therefore, we compared the differences in the adjacent regions and their adjacent genes between the 13 cp genomes (Figure 2). The genes *rps19*, *ndhF*, *ycf1*, and *trnH* were located in the junctions of LSC/IRb, IRb/SSC, SSC/IRa, and IRa/LSC, respectively. Compared to the relatively conserved location of *trnH* and *rps19*, the SSC/IR boundary regions were more variable. *trnH* in 11 cp genomes was between the SSC/IRb region with 2215 and 2216 bp in the SSC region and 7 and 8 bp in the IRb region, respectively. Specifically, *trnH* of NN9 (*S. wrightii*) was in the SSC region and *trnH* of NN15 extended 7 bp into the SSC region. In all 13 cp genomes, *ycf1* was located inthe SSC/IRa region, with 140–160 bp extending into the IRa region. No gene rearrangement or inversion events were observed.

Analysis of relative synonymous codon usage (RSCU) value showed that almost all amino acids had more than one synonymous codon, except for methionine and tryptophan (Figure 2A, Table S1). Nearly all protein-coding genes of the 13 cp genomes had a standard ATG start codon (RSCU = 1). Approximately half of the codons had an RSCU > 1, and most (29/31, 93.5%) ended with either A or T. RSCU = 1 of codon (TGG) indicated a balanced bias of the codon (Figure 2A). Leucine (2665, 10.82%) was the most common amino acid and cysteine (282, 1.14%) was the least common in the NN1 (*S. aethiopicum*) cp genome, as well as the other cp genomes (Figure 2B, Table S1).

### 3.2. Comparative Genome Analysis

Comparative analysis was performed on 13 cp genomes using KU682719 (*S. melongena* plastid) as a reference. The genomes exhibited a high degree of sequence synteny, suggesting a highly conserved evolutionary pattern. However, nucleotide substitutions, InDels, and length variations were observed among the species (Table 5). The number of SNPs in *S. melongena* (NN5, NN7, NN8) was smallest (9–10), with insignificant positions. The sample with the largest number of SNPs was *S. sisymbriifolium* (NN15; 259), which showed the mostdistinct genetic relationship with KU682719. *Solanum aethiopicum*, *S. indicum*, *S. macrocapon*, *S. wrightii*, and *S. torvum* showed moderate differences, with the first three closer to *S. melongena* and the latter two closer to *S. sisymbriifolium*; their ratios of intergenic SNPs were approximately 70%. Moreover, large differences were observed in the number, type, and positions of InDels among species, but little difference was observed among intraspecific species (Table 5). *Solanum melongena* contained the smallest number of InDels, ranging from 53 to 55. Eighty-five percent of the InDel loci were in the intergenic region, and more than 84% were insertions. The sample with the largest number of InDel loci was *S. sisymbriifolium* (NN15), which had the most distant relationship with KU682719 and largest number of differential InDel loci (up to 126). Ninety-six percent of the InDel loci were in the intergenic region, and 60.3% were insertions. *Solanum aethiopicum*, *S. indicum*, *S. macrocapon*, *S. wrightii*, and *S. torvum* were in the middle of the extremes. However, most InDels were in the intergenic region and the ratio of insertions in these five *Solanum* species

was closer to that of *S. sisymbriifolium* than to KU682719. Most SNPs and InDels were found in the LSC and SSC regions, not in the IR regions (Figure 2C,D). This demonstrated the conserved characteristics of the IR region. At the whole-genome level, there were limited subgenus differences in the InDels of the *Solanum* species. The most common subgenus difference was single-base insertions or deletions. The number of single-base InDels varied in *S. melongena* compared to that in other species, with only 2–3 in *S. melongena* and 38 in *S. sisymbriifolium*. In other close relatives of *Solanum*, single-base InDel mutations accounted for more than 30% of the total number of InDels. Furthermore, the average length of the InDels in the 13 samples was 6–9 bp (Figure 2E).

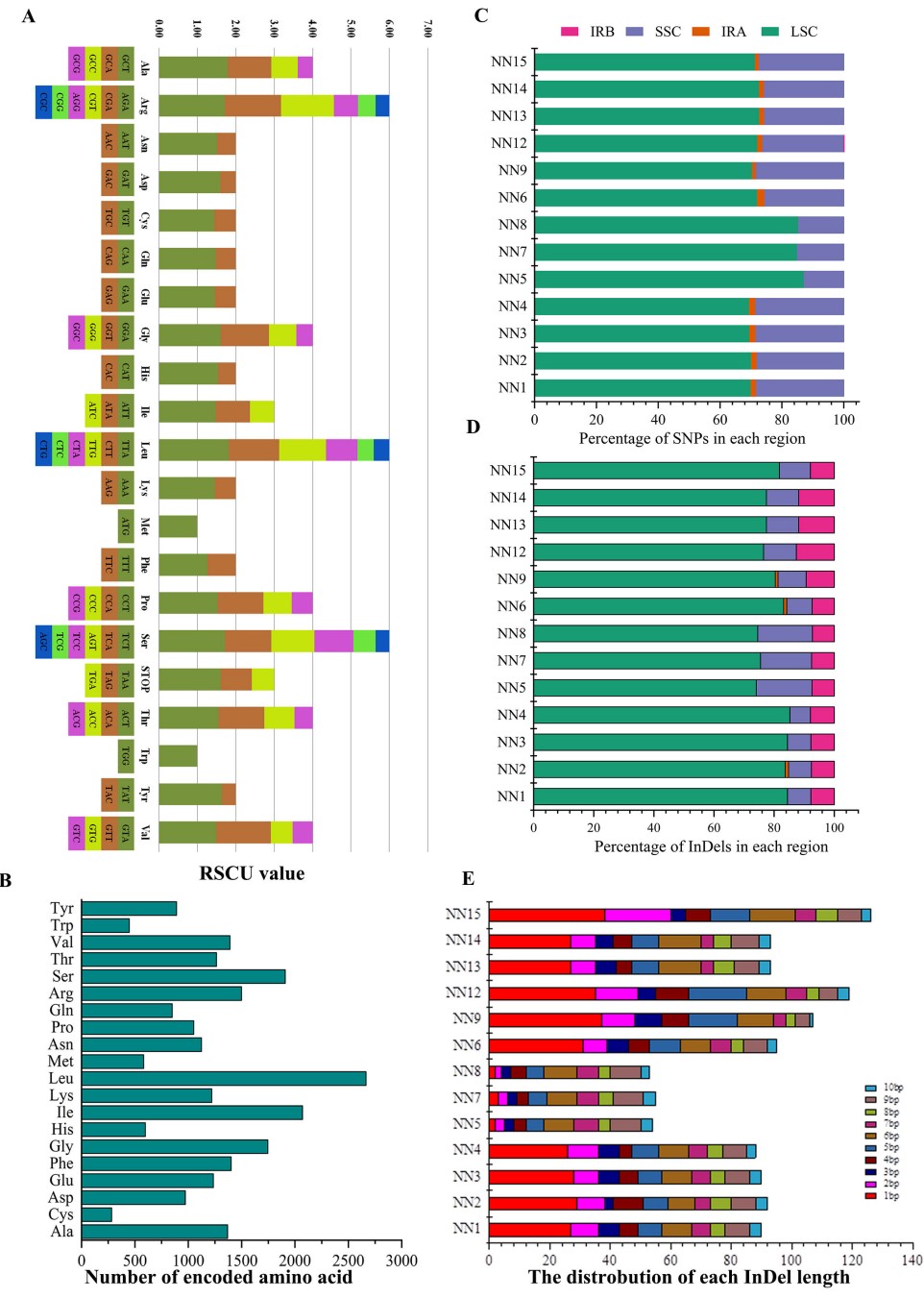

**Figure 2.** Codon, amino acid and sequence variation analyses of the cp genome. (**A**) Codon content and RSCU value of 20 amino acids and the stop codon of NN1 (*Solanum aethiopicum L.*) cp genome. (**B**) Numbers of encoded amino acids of NN1 cp genome. (**C**,**D**) Percentage of SNPs and InDels in each region. (**E**) The distribution of InDel length.

**Table 5.** InDel and SNP types of 13 cp genomes.

| Sample ID | Species | CDS InDel | Intergenic InDel | Insertion | Deletion | Total InDel | CDS SNPs | Intergenic_SNPs | ts/tv | Total_SNPs |
|---|---|---|---|---|---|---|---|---|---|---|
| NN1 | *S. aethiopicum L.* | 7 | 83 | 57 | 33 | 90 | 98 | 242 | 1.11 | 340 |
| NN2 | *S. aethiopicum L.* | 6 | 86 | 58 | 34 | 92 | 100 | 239 | 1.08 | 339 |
| NN3 | *S. aethiopicum L.* | 7 | 83 | 57 | 33 | 90 | 99 | 243 | 1.12 | 342 |
| NN4 | *S. aethiopicum L.* | 7 | 81 | 56 | 32 | 88 | 100 | 244 | 1.12 | 344 |
| NN5 | *S. melongena L.* | 8 | 46 | 46 | 8 | 54 | 9 | 30 | 0.44 | 39 |
| NN7 | *S. melongena L.* | 8 | 47 | 46 | 9 | 55 | 9 | 30 | 0.44 | 39 |
| NN8 | *S. melongena L.* | 8 | 45 | 45 | 8 | 53 | 10 | 31 | 0.44 | 41 |
| NN6 | *S. melongena L.* | 5 | 90 | 64 | 31 | 95 | 111 | 344 | 0.95 | 455 |
| NN9 | *S. wrightii L.* | 4 | 103 | 59 | 48 | 107 | 267 | 606 | 0.95 | 873 |
| NN12 | *S. torvum* | 5 | 114 | 72 | 47 | 119 | 243 | 599 | 0.91 | 842 |
| NN13 | *S. indicum L.* | 6 | 87 | 57 | 36 | 93 | 103 | 268 | 1.01 | 371 |
| NN14 | *S. indicum L.* | 6 | 87 | 57 | 36 | 93 | 103 | 267 | 1.01 | 370 |
| NN15 | *S. sisymbriifolium* | 5 | 121 | 76 | 50 | 126 | 259 | 637 | 0.94 | 896 |

*3.3. Repeat and SSR Analysis*

Four types of repeat sequences were determined: direct (forward), inverted (palindromic), complement, and reverse repeats. The numbers and distributions of the repeats in the 13 cp DNA were similar and conserved among these types. In total, 31–50 repeats were identified, including 16–30 forward repeats, 14–21 palindromic repeats, 0–9 reverse repeats, and 0–2 complement repeats (Figure 3A). Of the 13 cp genomes, forward and palindrome were the most abundant repeat types, with mostly consistent amounts. Reverse repeats were detected in *S. aethiopium* (NN1, NN2, NN3, NN4), *S. macrocapon* (NN6), *S. wrightii* (NN9), *S. indicum* (NN13), and *S. sisymbriifolium* (NN15) in amounts of 3, 3, 3, 3, 9, 5, 1, and 3, respectively (Figure 3B). Two complements were detected in NN1, NN2, NN3, and NN4 (Figure 3B). Most repeats were 30–39 bp, followed by 40–49 bp and over 50 bp (Figure 3B).

In the 13 cp genomes, 136–146 SSRs were identified: 107–121 mononucleotides (83.45%), 6–9 dinucleotides (5.52%), 4–7 trinucleotides (4.83%), 8–12 tetranucleotides (5.52%), and 1–3 pentanucleotides (0.69%) (Figure 3C,D). In NN1, 97.5% of the mononucleotides exhibited A or T types and 2.5% exhibited C or G types and all dinucleotides were composed of AT/TA (Figure 3E,F).

*3.4. Identification of the Most Variable Regions*

To determine the levels of sequence divergence, we calculated the nucleotide variability (*Pi*) values and investigated the levels of sequence divergence among genera in 46 *Solanum* genomes, including 13 *Solanum* cp genomes in this study and 33 other *Solanum* cp genomes downloaded from NCBI (Table S3). The *Pi* values within 300 bp across the 46 cp genomes ranged from 0 to 0.0695, with a mean value of 0.00351 (Figure 4A), indicating high similarity among species. However, hypervariable loci were also observed, including *rps16-CDS1_trnQ-UUG*, *atpH_atpI*, *ccsA*, *petA-psbJ*, *psbA*, *rpl32_trnL-UAG*, *ndhD*, *atpI*, *psbJ*, *rpl32*, *rps15_ycf1-D2*, and *rbcL_psaI* (*Pi* > 0.014). The *rps15_ycf1-D2*, *ndhD*, *ccsA*, *rpl32_trnL-UAG*, and *rpl32* loci were present in the SSC region; *psbJ*, *petA_psbJ*, *rbcL_psaI*, *atpI*, *atpH_atpI*, *rps16-CDS1_trnQ-UUG*, and *psbA* were present in the LSC region (Figure 4A). These species shared the same order and orientation of syntenic blocks, indicating no rearrangement in gene organization and illustrating that cp genomes tend to be conserved and collineated, particularly in the same plant family.

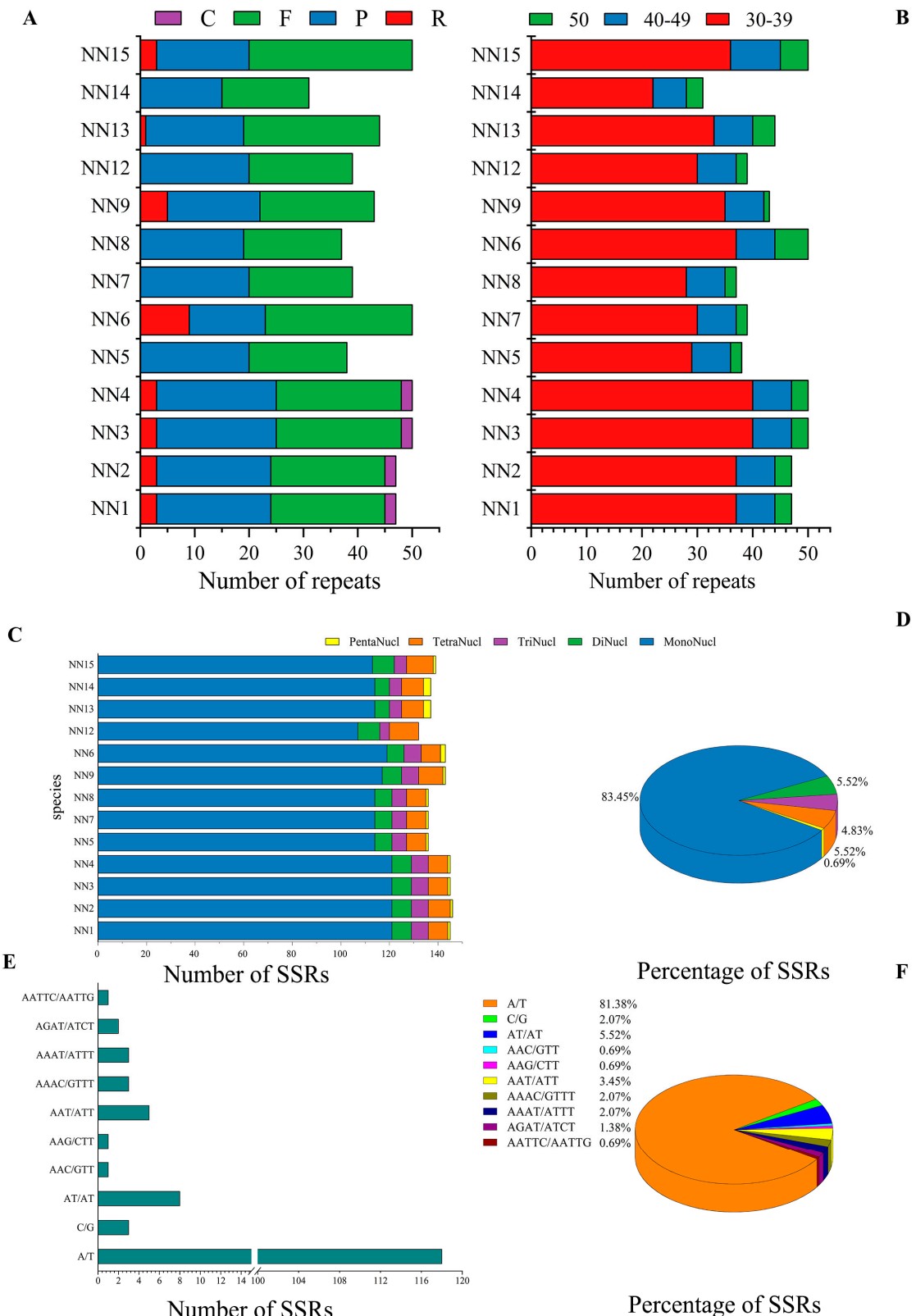

**Figure 3.** Repeat sequences and SSRs in 13 cp genomes. (**A**) F, P, R, and C indicate the repeat types F (forward), P (palindrome), R (reverse), and C (complement), respectively; (**B**) Repeats with different lengths. (**C**) Repeat unit and amounts of SSR in thirteen cp genomes. (**D**) Percentage of different SSR types in the NN1 cp genome. (**E**) SSR repeat sequences in the NN1 cp genome. (**F**) Percentage of repeated sequences in the NN1 cp genome.

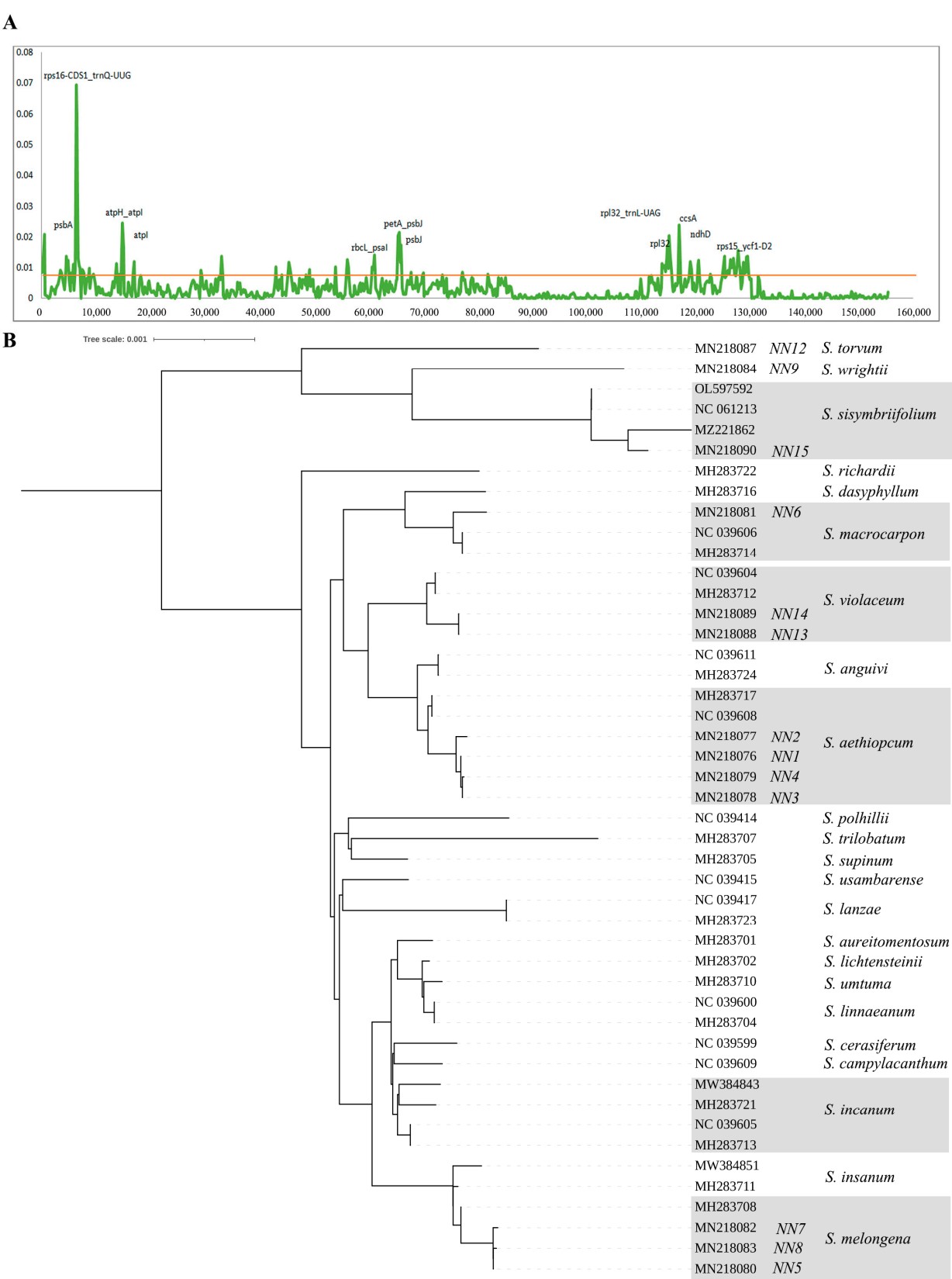

**Figure 4.** Variable regions and phylogenetic tree of the *Solanum* species. (**A**) Comparison of variable regions among 46 *Solanum* cp genomes. (**B**) Phylogenetic analysis of 46 *Solanum* cp genomes.

### 3.5. Phylogenetic Analyses

To better determine the phylogenetic position of *Leptostemonum* and further clarify the evolutionary relationships within *Leptostemonum*, phylogenetic analysis was performed based on the 13 cp genomes in this study and an additional 33 *Leptostemonum* cp genomes downloaded from NCBI (Table S3). A phylogenetic tree was constructed using the maximum-likelihood (ML) method. All these species formed two branches. One branch mainly consisted of the most economically important cultivated aubergine worldwide. The three cultivated aubergine species, the brinjal eggplant *(S. melongena)*, the scarlet eggplant (*S. aethiopicum*) and the gboma eggplant (*S. macrocarpon*), are closely related phylogenetically, and form a part of the subgenus *Leptostemonum*. Interestingly, high intraspecific genetic diversity was found among the three cultivated aubergine species, which formed different clades with their wild progenitors (Figure 4B). The African *S. anguivi* and the Asian *S. violaceum* are the sister species of *S. aethiopicum*. Moreover, three samples of *S. melongena* (NN5, NN7 and NN8) formed a phylogenetic tree with *S. melongena* (MH283708), but did not occur on the phylogenetic tree near *S. melongena* var. *insanum* (MH283711) (Figure 4B). The second branch consisted of *S. torvum*, *S. sisymbriifolium*, and *S. wrightii* (Figure 4B). *Solanum torvum*, *S. sisymbriifolium*, and *S. wrightii* are close relatives of aubergine, and are native to South America, Central America, and Africa. They are phylogenetically distant from the cultivated *S. melongena*.

## 4. Discussion

The *Solanum* subgenus *Leptostemonum* is the largest clade in genus *Solanum* and it contains three domesticated species (*Solanum melongena, S. aethiopicum and S. macrocarpon*). The taxonomy relationships and domestication of the cultivars and wild relatives is puzzling for breeders due to the large number of related species (Taher et al., 2017). In this study, we sequenced the cp genomes of seven subgenus *Leptostemonum* species, including three samples of *S. melongena*, four samples of *S. aethiopicum*, one sample of *S. macrocarpon* and five samples of their wild relatives, combining with 33 cp genome data of *Leptostemonum* species downloaded from NCBI to perform comparative genome and phylogenetic analysis. The results reveal precise intra- and interspecies relatedness within the subgenus *Leptostemonum*.

The cp genome of *Solanum melongena, S. macrocarpon, S. anguivi, S. aethiopicum, S. violaceum, S. torvum* and *S. sisymbriifolium* has been determined [18,39–41]. The cp genome of *S. wrightii* was the first report here. Chloroplast genomes have been successfully used in numerous phylogenetic studies of the subgenus *Leptostemonum* because of their high accuracy and resolution [18]. These analyses of cp genomes enhanced our understanding of the evolutionary trends and phylogenetic implications of *Leptostemonum*, and provide new molecular evidence for analyzing the genetic relationship of *Leptostemonum* species.

Interspecific hybridization is an effective way to create heterosis and species variation. Both *S. aethiopicum* and *S. macrocarpon* were partially interfertile to *Solanum melongena* [42,43]. *Solanum aethiopicum* exhibited higher cross-compatibility than *S. macrocarpon* when crossed with *S. melongena* [42,44]. This study strongly suggests that the genetic distances between *S. aethiopicum and S. macrocarpon* are smaller than that between *S. melongena*, respectively. *Solanum aethiopicum* consists of four cultivar groups (Gilo, Shum, Kumba, and Aculeatum), which exhibit high genetic diversity [45]. In our study, the four samples of *S. aethiopicum* were originally from different regions, and NN1 was the sister group of NN3 and NN4, while NN2 formed a split group. NN1, NN3 and NN4 were originally from Africa, whereas NN2 was from South America. This result was also present in the samples of *S. melongena*, which was comprised of four groups (E-H) [7,46]. These results suggested the genetic diversity of the subgenus *Leptostemonum* signatures associated with selection and domestication. *Solanum anguivi* is the ancestor of *Solanum aethiopicum*, which was consistent with the phylogenetic relationship of the two spices in this study [7,47,48]. *Solanum violaceum* occurs across Asia, but its close relatives were the African *S. anguivi* and *S. aethiopicum*, which suggested *S. violaceum* were originally from Africa [18]. The 2 samples of *S. violaceum*

(NN13 and NN14) occur on a different group from the *S. anguivi* (MH283724) in our study, implying that was misidentification of species or hybridization. *Solanum torvum*, *S. sisymbriifolium*, *and S. wrightii* are phylogenetically distant from the cultivated *S. melongena*. *S. torvum* can be crossed with *S. melongena* to produce sterility F$_1$, whereas *S. sisymbriifolium* cannot be crossed with *S. melongena* due to the distant genetic relationship [49,50]. There is no report about the interspecific hybridization between *S. wrightii* and *S. melongena*.

## 5. Conclusions

The cp genome sequences of 13 samples from seven *Solanum* species were analyzed in this study. Although the genomic structure and size were highly conserved, inverted repeat/single-copy boundary regions and variation among species were still detected, representing wide phylogenetic diversity in the genus. The availability of these cp genomes provides genetic information for identifying the species structure, taxonomy, phylogeny, and evolution in *Solanum*, along with insights into the utilization of *Solanum* plants.

**Supplementary Materials:** The following supporting information can be downloaded at https://www.mdpi.com/article/10.3390/cimb45040185/s1.

**Author Contributions:** Methodology, Q.Y. and P.W.; Software, Q.Y. and P.W.; Validation, Q.Y., L.C. and M.W.; Formal analysis, G.G.; Resources, C.X.; Data curation, Q.Y., W.L. (Weiliu Li) and Y.J.; Writing—original draft, Q.Y. and D.L.; Writing—review & editing, Y.L. and Y.W.; Supervision, W.L. (Wenjia Li) and R.C.; Project administration, Y.W.; Funding acquisition, Y.W. All authors have read and agreed to the published version of the manuscript.

**Funding:** This research was supported by the Science and Technology Planning Project of Guangxi (GuikeAD22035947, GuikeAA22068088-2), and Scientific Research Fund of Hunan Provincial Education Department (19B271).

**Institutional Review Board Statement:** Not applicable.

**Informed Consent Statement:** Not applicable.

**Data Availability Statement:** The data used in this study have been deposited in the NCBI Genbank with the accession number in Tables S1 and S3.

**Conflicts of Interest:** The authors declare no conflict of interest.

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
