# Peer review of "Characteristics, Comparative Analysis, and Phylogenetic Relationships of Chloroplast Genomes of Cultivars and Wild Relatives of Eggplant (Solanum melongena)"

_cimb, doi:10.3390/cimb45040185_

Round 1

Reviewer 1 Report (New Reviewer)

The presented manuscript is devoted to the study and characterization of the eggplant chloroplast genome. The results obtained are of high practical value. The manuscript deserves to be published. However, there are some minor remarks about the design of the manuscript.

1. references must be enclosed in brackets

2. 155 specify in more detail what model GTR+G was used?

3. -165 figure 1 in the text must be written in full, further in the text too

4. I recommend swapping tables 1 and 2.

5. in table 5, correct the inscriptions

6. There is a repetition in the Discussion section with the Introduction section.

7. add reference 46

8. The year of publication is given in bold after the name of the journal.

Author Response

Thank you very much for your comments and suggestions.  We have revised the manuscripts according to your suggestion. We accept  your suggestions except for point 4. We insist on using the original order of the table, because of the data in Table 2 that we uploaded to NCBI was from Table 1.

Reviewer 2 Report (New Reviewer)

In this study, the authors did  interspecific genetic and phylogenetic relationships analysis by sequencing and comparative analysis 13 complete chloroplast (cp) genomes of seven Solanum species combined with other 33 cp genomic sequences. The cp genomes data and phylogenetic analysis provides a clear molecular evidence for S. melongena and its wild relatives in origin and evolutionary relationships. and the precise intra- and inter-species relatedness within the subgenus Leptostemonum will provide positive effect on the breeding of eggplant improvement.

I have noticed a few edit mistakes:

line 58: there should a space between "functionsincluding".

line 112: need a space "DRAW v1.2.The cp genomes".

line 140: SSRS should be SSRs

line 141: "repeat 1 base 10 times or more; repeat 2 bases 6 times or more" should be: 1 base repeat 10 time or more; 2 bases repeat 6 times or more

line 165: "Ira, and IRb" should be "IRA, and IRB" to keep consistent.

line 308: (B) is missing.

line 325: "because on" should be because of.

line 336 and 338: "origin" should change to originally.

line 360: Patents: Participation

I suggest authors should check their typing again before publication. 

Author Response

Thank you very much for your comments and suggestions. We have revised the manuscript according to your suggestion.

Reviewer 3 Report (New Reviewer)

In my opinion, the manuscript entitled “Characteristics, comparative analysis, and phylogenetic relation-ships of chloroplast genomes of cultivars and wild relatives of egg-plant (Solanum melongena)” can be accepted in the present form. All applied methods are used according to high scientific standards. The obtained results would contribute to a better understanding of trichome formation.

Author Response

Thank you very much for your kindly comments and suggestions

This manuscript is a resubmission of an earlier submission. The following is a list of the peer review reports and author responses from that submission.

Round 1

Reviewer 1 Report

This work on plastid genomes of Solanum may be useful for those who specialize in genomics of Solanum and other Solanaceae.    I think, the article can be accepted for publication after the following concerns are met:
  1. Line 20. You mention "NN5" without explaining first to a reader what it is.
  2. Line 20. "Except for NN5, genes in twelve genomes were annotated". As far as I can see, NN5 was annotated too. Here it is https://www.ncbi.nlm.nih.gov/nuccore/MN218080 . Why did you write that it's not annotated?
  3. Line 29. "monophyletic group containing four clades". "Monophyletic group" is the same as "clade". These two terms have identical meaning. I think, this statement should be rephrased.
  4. Line 51. I don't understand the reason to shorten "chloroplast DNA" as "chloroplastDNA". I've read hundreds of articles on plastid genomes and have never seen such an abbreviation. Also, making an abbreviation just by removing a space is strange. Please, don't shorten "chloroplast DNA" as "chloroplastDNA".
  5. Lines 51-53. "The most effective and precise method for studying the phylogenetic position and diversity of a species is chloroplastDNA analysis". Actually, the use of the nuclear transcriptome or the nuclear genome results in more precise phylogenies, because their length is more than the length of the plastid genome. See, for example, https://pubmed.ncbi.nlm.nih.gov/31645766/ . The reason why scientists often use plastid genomes is because a) Usually plastid genome sequences are enough to calculate the phylogeny. b) It is cheaper to study the plastid genome that the transcriptome or the nuclear genome.
  6. Lines 62-63. "There is no interference by gene overlapping, deletions, and pseudogene". I don't understand what you mean here. For example, pseudogenes are sometimes found in plastid genomes.
  7. Line 94. "sequencing on the Sequel Sequencer". Were the PacBio reads made by the HiFi (also known as "CCS") or by the CLR technology? These two technologies utilized by PacBio provide reads with very different accuracy, so it's worth specifying.
  8. Line 100. "filling any gaps". Please, describe what methods you used to fill the gaps.
  9. Line 106. The link that you provide for BLAST is a wrong one. It is a brief description of BLAST which was written not by the creators of BLAST. The correct link should be to the article https://pubmed.ncbi.nlm.nih.gov/20003500/ .
  10. Lines 105-109. "A whole chloroplast Blast search... was performed... Gene Ontology". It's impossible to perform a BLAST search in Gene Ontology. With BLAST you can align sequences to sequences. However, the Gene Ontology database itself contains no sequences.
  11. Line 123. "BLAST 3.23". As far as I know, such a version of BLAST does not exist. The latest version is 2.13.0 ( https://ftp.ncbi.nlm.nih.gov/blast/executables/blast+/LATEST/ )
  12. Line 130. "Burrow-Wheeler alignment and SAM tools". The proper names of these programs are "BWA" and "SAMtools". See https://arxiv.org/abs/1303.3997 and https://pubmed.ncbi.nlm.nih.gov/33590861/ .
  13. You mentioned programs MISA, Mummer, Lastz, BWA, SAMtools, BioEdit, RepeatMasker, TRF but did not provide links to respective articles. Please, do this.
  14. In the Table 1 you have provided the information on Illumina reads, but no information on PacBio reads. Please, fix this problem.
  15. You're definitely not the first who sequenced plastid genomes of some of these species. For example, the plastid genome of Solanum melongena has already been deposited in GenBank in 2021, see https://www.ncbi.nlm.nih.gov/nuccore/MW384845.1 , and described in the article https://pubmed.ncbi.nlm.nih.gov/33964091/. I want you to clearly indicate in your article, plastid genomes of which of the studied 7 species were available before you. Also, please provide links to respective articles. Otherwise, a reader might overestimate the novelty of your work.
  16. Table 5. "NAD(P)H-dehydrogenase". The plastid NDH complex is not a NAD(P)H-dehydrogenase. It was earlies supposed to be a NAD(P)H-dehydrogenase, but this hypothesis was rejected. See https://pubmed.ncbi.nlm.nih.gov/26519774/ . It's better to call it "NADH dehydrogenase-like complex" or just "NDH complex".
  17. Line 202. "trnH in all 11 chloroplast genomes". You probably meant "13", not "11".
  18. The Figure 3B and the Figure 3C are identical. I think, it's better to replace them with one figure that have two horizontal axes. One horizontal axis with numbers and the other horizontal axis with frequencies. For example, one axis can be just below the other.
  19. You mention that rps16 is absent but don't explain what happened to it. Was it pseudogenised by a frameshift? Was this gene deleted completely? Please describe this somewhere is the article.
  20. Line 315. "Table S2". This link should be to Table S1. Also, the actual Table S2 (with the information about the codon usage) is not referenced anywhere in the text.
  21. Figure 8. I think it's worth mentioning that N13 and N14 (both of which belong to Solanum anguivi) didn't occur on the phylogenetic tree near the Solanum anguivi from the article https://pubmed.ncbi.nlm.nih.gov/30091787/ . That's interesting because it may indicate misidenfication of species or hybridization.
  22. Lines 339-340. "The genes infa and accD were lost in 13 species". To test this, I took the sequence of accD from Solanum melongena (https://www.ncbi.nlm.nih.gov/nuccore/MF818319.1) and aligned it by TBLASTN to the plastid genome of NN1. I see that it aligns entirely. Hence, I think that you made a mistake and accD may actually be present in some species. Results of DOGMA should be checked by aligning reference sequences by BLAST. Please, search for accD, infA and also for rps16 which has supposedly been lost in one species.
  23. Lines 340-341. "The frequent independent transfer of infa". Transfer to where?
  24. Line 353. "The ratio of ti/tv varied greatly among species". You didn't indicate in Materials and Methods how you calculated the transition/transversion ratio. Please, do this.
  25. Please, upload the sequencing reads to SRA. It is required by the rules of CIMB, see https://www.mdpi.com/journal/cimb/instructions#sequence .

Author Response

1.Thank you very much for your comments and suggestions. The writing here has been corrected.

2.It has been modified.

3.Thank you very much for your suggestions. It has been revised.

4.It has been revised.

5.According to your suggestions, the introduction was revised.

6.This sentence has been deleted.

7.This is a mistake.  The cp genome was reconstructed using a combination of de novo and reference-guided assemblies.

8.We used GapCloser to fill the majority of gaps  through local assembly.

9.It has been revised.

10.It have been revised in the paper.

11.It has been modified.

12.It has been revised.

13.They have been revised in the paper.

14.It has been fixed. See No.7

15.It is a good suggestion. We have rewritten this part in discussion. The cp genome of S. wrightii was the first report here.

16.Thank you for your suggestions. The writing here has been corrected.

17.tmH of NN9 (S. wrightii) was in the SSC region and trnH of NN15 extended 7 bp into the SSC region. 

18.The picture here has been modified.

19.The rps16 was absent from the cp genome of S.sisymbriifolium, resulting in a nonsense mutation. We have rewritten this part.

20.It has been revised.

21.Thank you for your good suggestions. We have rewritten this part.

22 and 23. We have rewritten this part.

24.We unable to provide the Methods calculated the transition/transversion ratio here, so we have abandoned this part of the data.

25.We will upload the sequencing reads  to SRA as soon as passible.

Reviewer 2 Report

The article is poorly designed, and structured in an unusual manner for this type of research topic. The material description, accession naming, and numbering are confusing. The lack of proper software nor citation of the second part of assembly gives bad aspects. Authors should read a similar study and follow it. Most of the species used in this analysis have been already published, and some of the published data is not included in the unorganized, unclearly labeled, and poorly highlighted phylogenetic tree. Much would be included when comparing such an important genus, but however, authors failed completely to highlight the important aspect of this comparison and majorly focused on SSR and tandem repeats with so many unnecessary figures on one hand, while many tables can be joined in one proper table for better representation of the data on the other. I strongly advise authors to provide the data availability information and confirm a precise inter and intra-specific comparison as they included different accessions from the same species, an opportunity to develop barcoding markers with much ease. The evolutionary analysis would become more important to focus on, to highlight the genes with the most dynamic variations with simple tests like ka/ks ratios rather than presenting percentages and overall diversity without specifying whether it is due to between or within species. Overall, it seems like the authors are not in control of their analyses and are not experienced enough to perform such important research on the Solanum genus. I strongly recommend the authors do a pan-plastome analysis, which is become widely applicable for inter and intra-specific comparisons at the genera level.

Author Response

 Thank you very much for your comments and suggestions. We have rewritten the manuscript according  to your suggestion.

Round 2

Reviewer 2 Report

I acknowledge the author's attempts to improve their outputs. However, rewriting the whole manuscript is not exactly what happened. Authors should provide point-by-point responses, be precise and justify each change they have made. They again skipped many of the advice suggestions and may require a serious revision following some examples that include inter- and intra-specific cp genomes. Please consider the pan-plastome approach, and define the differences among and between each studied species separately. The evolutionary parameters are important, especially since you have included wild accessions. The aim is unclear; the tree requires better visualization; please include the species name and put your code in brackets, arrange the branches and interpret the meaning of the work. Unless you provide precise responses to my comments, I will keep failing to track your improvements. Again, please follow a well-cited work with a similar design.

Author Response

Thank you for your comments and good suggestions. The manuscript has been marked up using the "Track changes" in Word, you can find each change we have made in our manuscript. Unfortunately, we are unable to provide  the results according to the pan-plastome approach in this study. We have made better visualization of the tree. 

Round 3

Reviewer 2 Report

Dear authors,

After several suggestions trying to improve your work, you have been working on modifying the text, rewriting the paragraphs aimlessly, and without clarifying the main points, I highlighted in the past two rounds. You keep writing in M&M one sentence for paragraphs and whole points regardless of my past suggestion to combine them.

Your study seems aimless; I don't see how cp sequencing will help Solanum breeding if such a thing exists. What output is expected from breeding eggplants x wild tomatoes!? a thing never recorded in nature.

Your extensive discussion talking one time about tomatoes and another time about eggplants is an evident confusion. If you can't handle a pan-plastome analysis at least separate the tomatoes from the eggplants in two separate analyses and compare them. Mixing all with these shallow objectives is very confusing. It gives no significant information knowing how many repeats, how many long repeats, and what types of repeats are found among diverse groups of Solanum genus.

Improve your material description, and add some photos as a supp material.

Please follow the journal instructions, the citing format is incorrect, and the tables should be positioned within the text, it is not convenient to keep up and down several times to revise your numbers!

Unless the work aim is clear and realistic and the analysis related to this aim and the results highlight it, I regret I will have my decision as it is.